

# Ice Crystal Characterization in Cirrus Clouds II: Radiometric Characterization of HaloCam for the Quantitative Analysis of Halo Displays

Linda Forster[1], Meinhard Seefeldner[1], Andreas Baumgartner[2], Tobias Kölling[1], and Bernhard Mayer[1,2]

[1]Ludwig-Maximilians-Universiät, München, Germany.
[2]Deutsches Zentrum für Luft- und Raumfahrt, Oberpfaffenhofen, Germany.

**Correspondence:** Linda Forster (Linda.Forster@physik.lmu.de)

**Abstract.** This study presents the weather-proof RGB camera HaloCam$_{RAW}$, which is part of the automated halo observation system HaloCam and designed for the quantitative analysis of halo displays. We present a procedure for both the geometric and absolute radiometric characterization of HaloCam$_{RAW}$ and demonstrate its application in a case study. The geometric calibration was performed using a chessboard pattern to estimate camera matrix and distortion coefficients. For the radiometric
characterization of HaloCam$_{RAW}$, dark signal and vignetting effect were determined to correct the measured signal. Furthermore, the spectral response of the RGB sensor and the linearity of its radiometric response were characterized. The absolute radiometric response was determined by cross-calibrating HaloCam$_{RAW}$ against the completely characterized specMACS imager. For a typical measurement signal the relative (absolute) radiometric uncertainty amounts to 2.8% (5.0%), 2.4% (5.8%), and 3.3% (11.8%) for the Red, Green, and Blue channel, respectively. The absolute radiometric uncertainty estimate is larger
mainly due to the inhomogeneity of the scene used for cross-calibration and the absolute radiometric uncertainty of specMACS. Geometric and radiometric characterization of HaloCam$_{RAW}$ were applied to a scene with a 22° halo observed on 21 April 2016. The observed radiance distribution and 22° halo ratio compared well with radiative transfer simulations assuming a range of ice crystal habits and surface roughness. This application demonstrates the potential of developing a retrieval method for ice crystal properties, such as ice crystal size, shape and surface roughness using calibrated HaloCam$_{RAW}$ observations together
with radiative transfer simulations.

## 1 Introduction

Halo displays are optical phenomena caused by the refraction and reflection of light by ice crystals in the atmosphere. Visible as bright and sometimes colorful circles and arcs, these optical displays appear in thin cirrus clouds or diamond dust (Wegener, 1925; Pernter and Exner, 1910; Minnaert, 1937; Tricker, 1970; Greenler, 1980; Tape, 1994; Tape and Moilanen, 2006). Halo
displays contain valuable information about ice crystal microphysical properties regarding their shape, surface roughness, and orientation (van Diedenhoven, 2014; Forster et al., 2017).

Probably the first attempt to retrieve information about ice crystal microphysical properties was reported by Lynch and Schwartz (1985). They used an image of a 22° halo taken with a Kodak Plus-X camera to infer ice crystal properties by com-



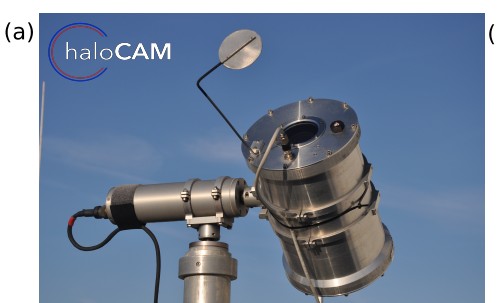
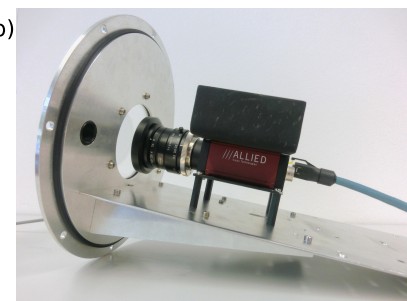

**Figure 1.** (a) HaloCam setup with HaloCam$_{\text{RAW}}$. The circular shade blocks direct sunlight and is covered with black, anti-reflective color on the side facing the camera. (b) Interior of HaloCam$_{\text{RAW}}$'s weather-proof casing. HaloCam$_{\text{RAW}}$ consists of an Allied Vision Manta G-235C camera with a Kowa lens of $6\,\text{mm}$ focal length. The camera is fixed on a "drawer", which is attached to the circular front lid of the cylindric weather-proof casing. A metal sheet is glued on the top of the camera body with a thermal compound to support the cooling of the camera. For the window of the camera casing, a Heliopan UV filter with anti-reflection coating was used. The PoE (Power over Ethernet) cable is guided inside the camera casing via a water-proof connecting plug through the lid just below the window.

paring their observations qualitatively with scattering phase functions. The calibration of this camera was performed by taking pictures of calibrated intensity wedges and grids with several exposures. However, no further details on the calibration method or its application were provided. Recently, Dandini et al. (2018) presented a geometrically calibrated all-sky camera with an equisolid angle projection. The camera was equipped with a sun-tracking shadow disk for the observation of halo displays.

Dandini et al. (2018) determined the center pixel coordinates and the azimuthal deviation from true north by comparing star and planet trajectories with their theoretical angular positions.

So far, camera observations of halo displays only used geometrically calibrated relative intensity measurements to retrieve information about ice crystal properties. However, observations of halo displays can not be directly linked to these properties due to the effect of multiple scattering and the contribution of aerosol below the cirrus cloud (Forster et al., 2017). To disen-

tangle these effects and to retrieve ice crystal microphysical properties, the observations have to be compared with radiative transfer simulations. For such a comparison calibrated measurements of sky radiance are required and thus the camera has to be characterized both geometrically and radiometrically.

For other applications, methods have been presented e.g. for the relative radiometric calibration of camera systems used for solar energy forecasting (Urquhart et al., 2015, 2016) and for the absolute radiometric characterization of the hyper-spectral

imager specMACS (Ewald et al., 2016), which is used in this study. For a detailed review of camera calibration methods tailored to remote sensing of the atmosphere see Ewald et al. (2016) and references therein.

This paper presents HaloCam$_{\text{RAW}}$ a sun-tracking camera system for the radiometric analysis of halo displays. Methods for geometric as well as radiometric calibration of this camera system will be presented allowing for a quantitative analysis of halo observations. The geometric calibration of HaloCam$_{\text{RAW}}$ was performed with the "chessboard method" (Zhang, 2000;

Heikkilä and Silvén, 1997) as described in Forster et al. (2017). The radiometric calibration is partly inspired by the procedure





and notation for the specMACS instrument (Ewald et al., 2016) which serves as a reference for the absolute radiometric calibration of HaloCam$_\text{RAW}$.

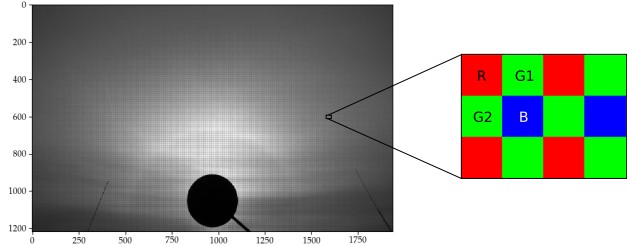

**Figure 2.** HaloCam$_\text{RAW}$ image as detected by the sensor. The image exhibits the superimposed mosaic pattern of the Bayer color filter array (CFA, Bayer (1975)) which is shown schematically on the right with the red (R), the 2 green (G1, G2), and the blue (B) channels.

## 2 HaloCam system description

HaloCam, the weather-proof and sun-tracking camera system allows for an automated observation of halo displays (Forster

et al., 2017). The camera system was developed at the Meteorological Institute Munich (MIM) of the Ludwig-Maximilians University (LMU) and installed on the rooftop platform for continuous observations. HaloCam consists of two wide-angle cameras which are mounted on a sun-tracking system as shown in Fig. 1. While the term HaloCam refers to the camera system including the sun-tracking mount and both cameras, the terms HaloCam$_\text{JPG}$ and HaloCam$_\text{RAW}$ represent the specific cameras. Due to on-chip post-processing and JPEG compression, HaloCam$_\text{JPG}$, which was presented in Forster et al. (2017), cannot be

used for a quantitative analysis and will not further be discussed in the following.

This paper focuses on HaloCam$_\text{RAW}$, which provides the "raw", i.e. unprocessed and uncompressed, signal from the sensor. HaloCam$_\text{RAW}$, as shown in Fig. 1, is an Allied Vision Manta G-235C camera equipped with a Kowa LM6HC wide-angle lens with 6 mm focal length (cf. Table 1). With its 1/1.2" Sony IMX174 CMOS sensor, HaloCam$_\text{RAW}$ yields a field of view (FOV) of 87° in the horizontal and 65° in the vertical direction[1]. The RGB sensor features 1936×1216 pixels and captures spectral

information via so-called primary color (red, green, blue) filters. These are located over the individual pixels, arranged in a Bayer color filter array (CFA) (Bayer, 1975). For this study, HaloCam$_\text{RAW}$ is used in "raw" mode, i.e. the signal measured by the camera sensor is directly used without (color) processing. This provides monochrome images with superimposed Bayer checkerboard pattern as shown in Fig. 2. A schematic illustration of the Bayer CFA layout with the red (R), blue (B) and two green (G1, G2) channels is displayed as a magnified detail of Fig. 2.

Since HaloCam$_\text{RAW}$ itself is not weather-proof, a protecting aluminum casing was built (cf. Fig. 1). The casing has a cylindric shape and the camera is fixed on a "drawer" which is attached to the circular front lid (cf. Fig. 1b). An anti-reflection

---

[1]The FOVs provided here are the result of the geometric calibration in Section 3.





coated UV filter (Heliopan GmbH) is used as a window for the casing. The PoE (Power over Ethernet) cable is guided inside the casing through the circular lid just below the window via a water-proof connecting plug. $\text{HaloCam}_{\text{RAW}}$ is operated in an automatic exposure mode. It measures the histogram of the current image to adjust the exposure time of the next image so that bright areas are not saturated.

The HaloCam camera system, with its two camera $\text{HaloCam}_{\text{JPG}}$ and $\text{HaloCam}_{\text{RAW}}$, is operated by the same sun-tracking mount described in Forster et al. (2017). The mount features two stepper motors with gear boxes to automatically align the position of the camera with the calculated direction of the sun. The stepper motors allow an incremental positioning of $2.16\,\text{arcmin}$ per step (Seefeldner et al., 2004) and have an estimated pointing accuracy of about $\pm0.5°$ ($2\sigma$ confidence interval) (Forster et al., 2017). Every $10\,\text{s}$ HaloCam's position relative to the sun is updated and a picture is recorded. Using a sun-tracking

mount is ideal for the automated observation of halo displays and later image processing since it ensures that the center of the camera is aligned with the sun and thus all recorded halo displays are centered on the images. To protect the lens from direct solar radiation and to avoid stray light a small circular shade fixed in front of the camera is sufficient with this setup (cf. Fig. 1). The camera FOV and the sensor resolution were chosen to achieve the optimal trade-off between a large coverage of the sky with high spatial resolution and low image distortion. 22° halo, sundogs, upper and lower tangent arc and circumscribed

halo, which are the most frequently observed halo displays according to Sassen et al. (2003) and Arbeitskreis Meteore e.V. Sektion Halobeobachtungen (AKM, https://www.meteoros.de), are captured by $\text{HaloCam}_{\text{RAW}}$'s FOV. By capturing the most frequently observed halo displays $\text{HaloCam}_{\text{RAW}}$ is expected to provide sufficient information to gain a better understanding of the relationship between halo displays and typical ice crystal properties in cirrus clouds.

    Since the presence or absence of the rare 46° halo might provide additional information $\text{HaloCam}_{\text{RAW}}$ was tilted upward

by 26° compared to $\text{HaloCam}_{\text{JPG}}$. This setup allows to observe the upper part of both the 22° and 46° halo (cf. Fig. 2). The upper part of these halo displays is usually brighter and more frequently visible than the lower part due to a shorter optical path through the atmosphere and less multiple scattering which acts to diminish the brightness contrast of the halo display (e.g. Gedzelman (2011); Forster et al. (2017)).

    The HaloCam system was installed in September 2013 on the rooftop platform of MIM (LMU) in Munich with $\text{HaloCam}_{\text{JPG}}$

only and was extended in September 2015 by $\text{HaloCam}_{\text{RAW}}$. The MIM rooftop platform hosts a cloudnet site (Illingworth et al., 2007) featuring operational measurements by a MIRA-35 cloud radar (Görsdorf et al., 2015), a CHM15kx ceilometer (Wiegner et al., 2014), and a RPG-HATPRO microwave radiometer. Further operational measurements are performed by a CIMEL sun photometer, which is part of the AERONET (Aerosol Robotic Network) network (Holben et al., 1998) as well as with the institute's own sun photometer SSARA (Sun–Sky Automatic Radiometer, Toledano et al. (2009, 2011)). HaloCam

observations ideally complement these measurements to retrieve information about ice crystal properties.

## 3    Geometric calibration

Halo displays are single scattering phenomena. Thus, their relative position to the sun is directly linked to the scattering phase function of the ice crystals producing them: the phase function of smooth hexagonal solid columns, for example, predicts a





**Table 1.** HaloCam$_{\mathrm{RAW}}$ specifications

| Lens | Kowa LM6HC |
|---|---|
| Focal length | 6 mm |
| Aperture | F1.8 - F16.0 (manual) |
| Horizontal field of view | 87° |
| Vertical field of view | 65° |
| Camera | Allied Vision, Manta G-235C |
| Interface | IEEE 802.3af (PoE) |
| Protection class | None |
| Operating (ambient) temperature | 5 °C to 45 °C |
| Sensor | 1/1.2" CMOS, RGB |
| | Sony IMX174LQJ |
| Maximum bit depth | 12 bit |
| Sensor resolution | 2.4 MPixel |
| Sensor pixels | 1936×1216 |
| Shutter type | global shutter |
| Image formats | Bayer (8 or 12 bit) |
| | Mono (8 or 12 bit) |
| | RGB (8 bit), YUV |
| Measures $w \times h \times d$ | $86.4 \times 44 \times 29$ mm |
| Weight (camera body + lens) | 400 g + 215 g |

22° and 46° halo at a scattering angle ($\Theta$) of 22° and 46°, respectively (e.g. Yang et al. (2013)). To uniquely identify halo displays in terms of their relative position to the sun and to allow a quantitative analysis of HaloCam$_{\mathrm{RAW}}$ images a geometric calibration is necessary which determines the transformation between the image pixels and scattering angles. For the geometric calibration of HaloCam$_{\mathrm{RAW}}$ several pictures are taken of a chessboard pattern from different angles and orientations. These

5  pictures are used to estimate the intrinsic camera parameters as well as the radial and tangential distortion parameters of the lens. This method was already used in Forster et al. (2017). It is based on Zhang (2000) and Heikkilä and Silvén (1997) and was implemented in OpenCV by Itseez (2015) with a detailed reference in Bradski and Kaehler (2008).

Using the distortion coefficients and intrinsic parameters, the camera pixels can be undistorted and mapped to the spherical world coordinate system. Projected onto the image plane, a zenith ($\vartheta$) and azimuth angle ($\varphi$) can be assigned to each pixel

10  relative to the center of the sun. In this case the relative zenith angle $\vartheta$ corresponds to the scattering angle $\Theta$.

An overlay of the relative zenith ($\vartheta$) and azimuth ($\varphi$) for the HaloCam$_{\mathrm{RAW}}$ red channel (R-channel) is displayed in Fig. 3 with representative contour lines at $\vartheta = 22°$, 35° and 46°. The geometric calibration was performed for the raw image (cf.





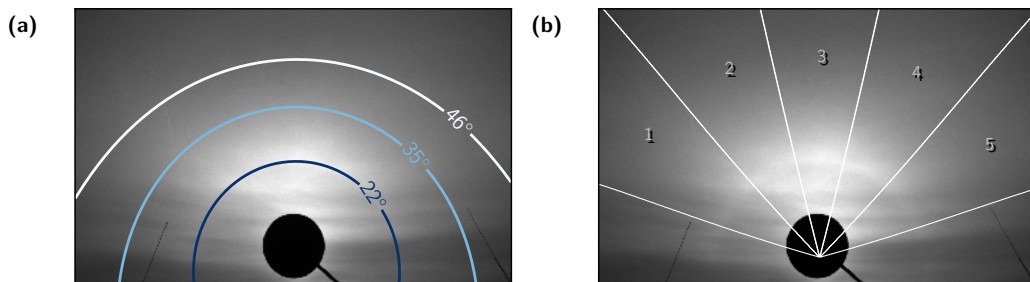

**Figure 3.** (a) HaloCam$_{\text{RAW}}$ image (red channel) from 2 February 2016, 9:42 UTC with corresponding scattering angle ($\vartheta$) grid and representative contour lines at $22°$, $35°$ and $46°$. (b) Relative azimuth angle ($\varphi$) grid with numbered labels for the 5 image segments, each spanning an interval of $30°$.

Fig. 2). The horizontal and vertical FOV of HaloCam$_{\text{RAW}}$ can be estimated from the calculated scattering angle grid to $\sim87°$ and $\sim65°$, respectively. With a resolution of $608{\times}968$ quadratic pixels the angular resolution of each of the 4 color channels amounts to about $0.1°$. Note that the individual channels are extracted from the Bayer pattern without interpolation (cf. Fig. 2). The HaloCam$_{\text{RAW}}$ image is separated into segments using the relative azimuth angle $\varphi$. Figure 3b indicates the 5 azimuth

segments, each $30°$ wide. For further analysis the radiance within each image segment is averaged in the azimuthal direction ($\varphi$), i.e. along the $22°$ halo. The angular width of the segments allows to reduce measurement noise and at the same time maintains the necessary spatial resolution to resolve brightness fluctuations of the $22°$ halo. Upper tangent arcs would be covered by segment no. 3, while sundogs would be visible below segments no. 1 and 5. Thus, if halo displays are visible, segments no. 1, 2, 4, and 5 are expected to contain only features of the $22°$ and $46°$ halo. Tilting the camera upward by $26°$, as

shown in Fig. 3a allows to observe of the more frequent upper part of the $22°$ and $46°$ halo and is therefore more suitable for a quantitative analysis.

## 4   Radiometric characterization

Each sensor pixel is a semiconductor device which converts light into electrical charge and can be treated as an independent radiometric sensor. The charge collected on a pixel is converted to a voltage and then to a digital value by the A/D converters,

which introduces noise at each step. The signal measured by the sensor can be expressed as

$$S = S_0 + S_{\text{d}} + \mathcal{N} \tag{1}$$

with $S_{\text{d}}$ the dark signal, $S_0$ the radiometric signal, and the measurement noise $\mathcal{N}$, as defined in Ewald et al. (2016). The measurement noise $\mathcal{N}$ is the sum of the radiometric signal noise $\mathcal{N}_0$ and the dark signal noise $\mathcal{N}_{\text{d}}$

$$\mathcal{N} = \mathcal{N}_0 + \mathcal{N}_{\text{d}}. \tag{2}$$





In the following sections the components of the measured signal $S$ will be characterized and their sensitivity on the camera settings and ambient conditions will be investigated. The dark signal measurements were performed in the optics laboratory of the Meteorological Institute at LMU on 16 July 2015. The measurements at the Large Integrating Sphere (LIS) and the spectral characterization of the sensor were performed at the Calibration Home Base (CHB) (Gege et al., 2009; DLR Remote

Sensing Technology Institute, 2016) of the Remote Sensing Technology Institute at the German Aerospace Center (DLR) in Oberpfaffenhofen on 28 June 2016. In the subsequent sections, temporally averaged values are indicated by angle brackets while spatial averages are denoted by an overbar. If not stated explicitly all variables are defined pixel-wise and uncertainties are provided with a $1\sigma$ confidence interval.

### 4.1 Dark signal

The dark signal $S_d$ is defined as the signal which can be measured when no light is entering the camera, i.e. the shutter is closed. This implies $S_0 = 0$ and Eq. (1) becomes

$$S = S_d + \mathcal{N}_d. \tag{3}$$

For an averaged dark image $\langle S \rangle$ the remaining noise approaches zero $\langle \mathcal{N}_d \rangle \to 0$ and the dark signal $S_d$ can directly be measured. The dark signal consists of the dark current $s_{dc}$, which is caused by thermally generated electrons and holes within the

semiconductor material of the sensor, and the read-out offset of the A/D converters $S_{read}$. The dark current $s_{dc}$ depends on the temperature $T$ and the exposure time $t_{expos}$

$$S_d(T) = s_{dc}(T)\, t_{expos} + S_{read}. \tag{4}$$

Thermal electrons are generated randomly over time with an increasing rate as the temperature rises. Since HaloCam$_{RAW}$ has no external shutter, the dark signal during operation has to be estimated from the laboratory characterization. The following

experiments were performed in a dark room and the camera lens was covered with an opaque cloth.

Figure 4 displays the dark signal $\langle S_d \rangle$ averaged over 100 images for an exposure time of $t_{expos} = 2.0\,\mathrm{ms}$ and a device temperature of $45\,°\mathrm{C}$ for the R-channel. The temporally (over 100 images) and spatially (over all pixels) averaged dark signal amounts to about $\overline{\langle S_d \rangle} = (16.7 \pm 0.2)\,\mathrm{DN}$ (Digital Number). For this number of averaged images, the dark signal in Fig. 4 does not show a significant spatial pattern. The same is true for the G1-, G2-, and B-channel. Therefore, a spatially averaged dark

signal will be used for the following analysis and later image processing. Figure 5 shows the dependency of the dark signal on exposure time for a constant temperature inside the camera of $45\,°\mathrm{C}$. In operational mode and under daylight conditions typical exposure times of 1 to $3\,\mathrm{ms}$ are used. For exposure times up to $50\,\mathrm{ms}$ the mean dark signal amounts to about 16.7 DN with a standard deviation of 0.8 DN for the R-channel. The standard deviation of the mean dark signal for exposure times below $50\,\mathrm{ms}$ is less than $0.02\,\mathrm{DN}$ or $0.1\,\%$. As observed by Urquhart et al. (2015) and Ewald et al. (2016) (VNIR camera of specMACS),

the dark signal appears to be independent of the exposure time. For larger exposure times, which are shaded in gray in Fig. 5, the dark signal as well as its standard deviation increase slightly. This behavior is most likely a combination of the increasing dark current signal due to a longer exposure time and an increase of the read noise signal $S_{read}$ caused by the A/D converters.



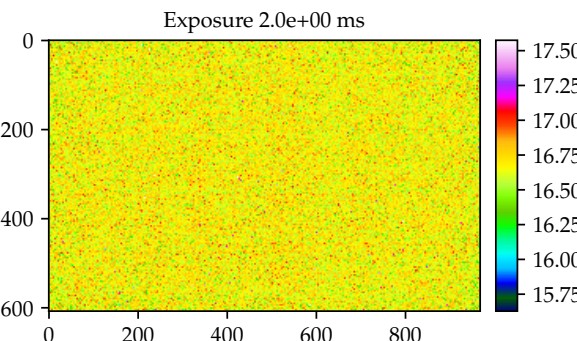

**Figure 4.** HaloCam$_{\mathrm{RAW}}$ dark signal (in DN) of the R-channel, averaged over 100 images. An exposure time of $t_{\mathrm{expos}} = 2.0\,\mathrm{ms}$ was chosen and a temperature of $45\,^{\circ}\mathrm{C}$ was measured inside the camera.

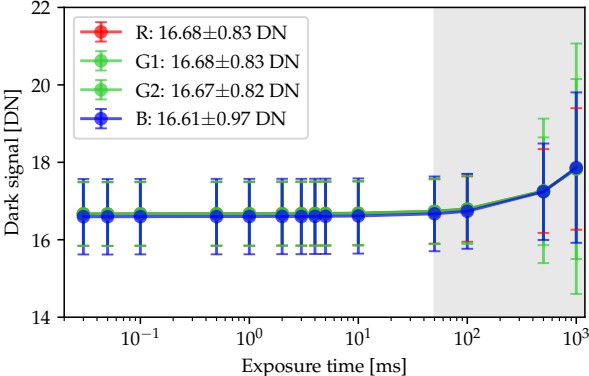

**Figure 5.** HaloCam$_{\mathrm{RAW}}$ dark signal and dark signal noise of all four channels (R, G1, G2, B) for different exposure times ranging from $0.03\,\mathrm{ms}$ to $1000\,\mathrm{ms}$. The camera's internal temperature was constant at $45\,^{\circ}\mathrm{C}$. The dark signal average and the standard deviation were evaluated over 100 images for each exposure time. The values in the legend represent the average and standard deviation of the dark signal over all measurements and exposure times for the respective channel.

To investigate the temperature sensitivity of the dark signal, measurements were performed with the camera set up inside a climate chamber (Weiss[2], SB11/160/40) in a dark room and with the camera lens covered. The temperature inside the climate chamber can be adjusted between $-40\,^{\circ}\mathrm{C}$ to $180\,^{\circ}\mathrm{C}$ with increments of $0.1\,^{\circ}\mathrm{C}$. The estimated accuracy is about $0.05\,\mathrm{K}$. For the dark measurements with HaloCam$_{\mathrm{RAW}}$ the temperature was increased from $10\,^{\circ}\mathrm{C}$ to $50\,^{\circ}\mathrm{C}$ in steps of $5\,^{\circ}\mathrm{C}$. Within this temperature range the averaged dark signal varied less than $0.5\,\mathrm{DN}$. To obtain an estimate for the temporal drift of the dark

---

[2]Weiss Klimatechnik GmbH, Greizer Straße 41-49, D-35447 Reiskirchen-Lindenstruth





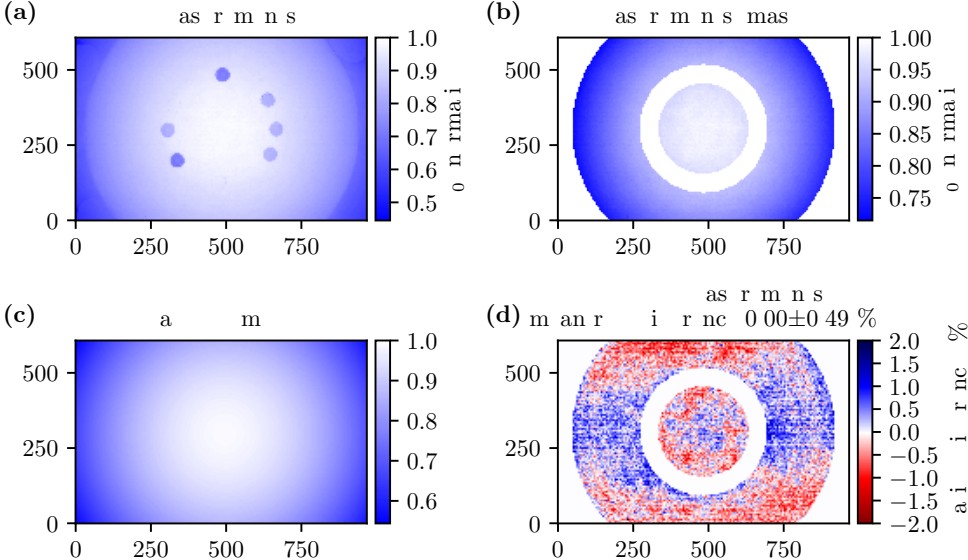

**Figure 6.** (a) HaloCam$_{\mathrm{RAW}}$ dark signal corrected measurements $S_0$ (R-channel), which are normalized to 1, of the large integrating sphere (LIS) averaged over 6 different camera orientations, 40 images each. (b) $S_0$, normalized to 1, as in (a) with a mask applied to the areas where the holes and the edge of the LIS are visible. (c) Flat-field model for HaloCam$_{\mathrm{RAW}}$ R-channel fitted against the measurements with a 2-dimensional 2nd order polynomial. (d) Relative difference between flat-field model and masked measurements.

signal, the mean standard deviation was calculated using all recorded dark images for the different camera temperatures and exposure times and results to less than $2.20\,\mathrm{DN}$ for the four channels. For the dark signal correction of the HaloCam$_{\mathrm{RAW}}$ data, the respective mean values from Fig. 5 are used for each of the four channels: $16.68\,\mathrm{DN}$, $16.68\,\mathrm{DN}$, $16.67\,\mathrm{DN}$, and $16.61\,\mathrm{DN}$ for the R-, G1-, G2-, and B-channel, respectively.

## 4.2 Vignetting correction

The wide-angle lens of HaloCam$_{\mathrm{RAW}}$ causes a decreasing radiometric signal $S_0$ for raypaths further away from the optical axis of the lens. This illumination falloff towards the edges of the sensor is called vignetting. There are two different types of vignetting:

1. Optical vignetting occurs when the ray bundle, which forms the image, is truncated by two or more physical structures in different planes (Bass et al., 2010). Typically, one is the nominal aperture and another is the edge of a (multiple element) lens. This kind of vignetting naturally occurs in all lenses and typically affects peripheral light rays, far off the optical axis.

2. Natural vignetting describes the effect that for off-axis image points the illumination is usually lower than for the image point on the optical axis (Bass et al., 2010).



Optical vignetting can be diminished by reducing the entrance pupil, i.e. the aperture by increasing the $f$-number. According to Bass et al. (2010) the $f$-number is defined by

$$f\text{-number} = \frac{\text{focal length}}{\text{entrance pupil diameter}}. \tag{5}$$

The $f$-number of HaloCam$_{\mathrm{RAW}}$'s Kowa lens can be adjusted mechanically between 1.8 and 11 by a screw. A fixed value of $f$-number = 8 was chosen for all measurements and the calibration. For the observation of halo displays close to the sun this represents a good trade-off between a small aperture and short exposure times.

To obtain a model for the non-uniformity of the sensor response as a function of the pixel location, flat-field measurements were performed using the large integrating sphere (LIS) at CHB as a uniform light source. Several measurements were performed with the same exposure time. To minimize the impact of inhomogeneities in the brightness of the integrating sphere, images were recorded at 6 different orientations by rotating the camera around its own axis, i.e. with the center of the camera roughly pointing to the center of the sphere. For each orientation 40 images were recorded, dark signal corrected and averaged. The measurements, which were averaged over the rotation angles of the camera relative to the sphere, are shown in Fig. 6a with the signal normalized to 1. The spherical patches visible in the figure are due to a hole in the sphere, which allows for injecting a laser as light source for specific experiments. The hole appears at different locations on the image due to the different rotation angles of the camera. Owing to the large field of view of the camera, the edge of the two hemispheric components of the LIS is visible. In order to fit a model to the flat-field measurements, these two regions were masked out as displayed in Fig. 6b. The flat-field model correcting for the vignetting effect was determined by fitting a 2-dimensional (2D) second order polynomial to the averaged and masked measurements.

$$F = a \cdot r^2 + b \cdot r + c, \tag{6}$$

where $r^2 = |\mathbf{x} - \mathbf{x_0}|^2$ is the distance of the location $\mathbf{x}$ from the image center $\mathbf{x_0}$ in pixel units. The result is depicted in Fig. 6c for the R-channel with the following parameterization:

$$F = -1.23 \times 10^{-6} \cdot r^2 - 4.30 \times 10^{-5} \cdot r + 0.99, \tag{7}$$

with $y_0 = 297.2$ and $x_0 = 473.8$. Finally, Fig. 6d shows the relative difference between the flat-field model and the measurements in percent averaged over all 6×40 images. The fluctuations of the signal difference are due to inhomogeneities of the integrating sphere. However, these inhomogeneities are not relevant for the image processing procedure since the flat-field model is used to correct the camera measurements. The average difference between model and measurement amounts to $(0.0 \pm 0.5)\,\%$ for the R-channel with similar values for the remaining channels. Correcting for the vignetting the flat-field corrected signal $S_{\mathrm{F}}$ is defined by

$$S_{\mathrm{F}} = S_0 / F, \tag{8}$$

with the radiometric signal $S_0$ and the flat-field correction $F$. This correction is applied to the radiometric signal $S_0$ of the red, green and blue channel separately.





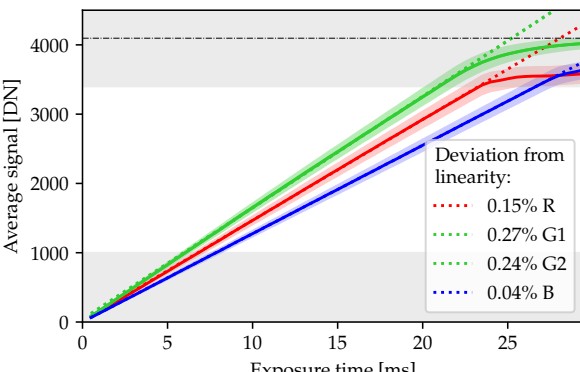

**Figure 7.** HaloCam$_{\mathrm{RAW}}$ average radiometric signal $S_0$ as a function of exposure time $t_{\mathrm{expos}}$ for the four channels. In operational mode, the measured signal typically ranges between $1000$ and $3400\,\mathrm{DN}$, where the averaged signal deviates from a linear behavior between $0.04\%$ for the B-channel and $0.27\%$ for the G1-channel. Signals below and above this range are shaded in gray. For signals close to saturation ($4095\,\mathrm{DN}$, black dashed line) the signal deviates clearly from a linear behavior.

## 4.3 Linearity of radiometric response

Similar to Ewald et al. (2016) the linearity of the CMOS sensor of HaloCam$_{\mathrm{RAW}}$ was investigated by measuring a temporally stable light source using different exposure times. This experiment was performed using the LIS at CHB. Baumgartner (2013) characterized the output stability of the LIS to better than $\sigma = 0.02\%$ over a time range of $330\,\mathrm{s}$. For a perfectly linear sensor with response $R$, the photoelectric signal $\widetilde{S_0}$ should increase linearly with exposure time $t_{\mathrm{expos}}$ and radiance $L$

$$\widetilde{S_0} = R L t_{\mathrm{expos}} = s_{\mathrm{n}} t_{\mathrm{expos}}\,, \tag{9}$$

with the normalized signal $s_{\mathrm{n}}$ defined by

$$s_{\mathrm{n}} = R L\,. \tag{10}$$

The deviation of the actually observed signal $S_0$ from the linear relationship of $\widetilde{S_0}$ is called photo response non-linearity. The actually observed signal $S_0$ can be written as

$$S_0 = F s_{\mathrm{n}} t_{\mathrm{expos}} = F R L t_{\mathrm{expos}}\,, \tag{11}$$

with the flat-field correction $F$ and it follows that the normalized signal can be obtained by

$$s_{\mathrm{n}} = S_0 / (F t_{\mathrm{expos}})\,. \tag{12}$$

Figure 7 shows the measured radiometric signal $S_0$ for exposure times $t_{\mathrm{expos}}$ ranging from $0.5\,\mathrm{ms}$ to $29.5\,\mathrm{ms}$, averaged over 5 images for each exposure time. For exposure times larger than $23\,\mathrm{ms}$ some pixels start to get overexposed. These pixels are





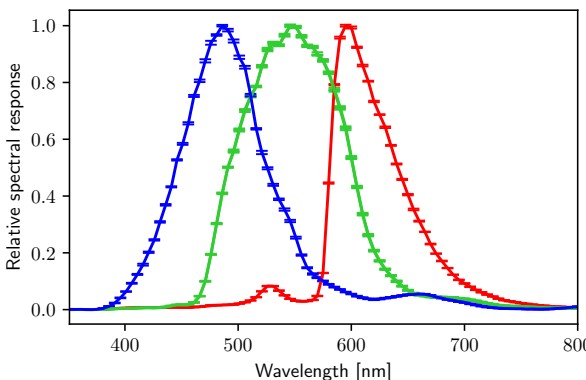

**Figure 8.** HaloCam$_\mathrm{RAW}$ relative spectral response for the R-, G1-, G2-, and B-channel.

excluded from the analysis. The measured signal ranges between about 84 and 4020 DN for the G1- and G2-channels. For the data analysis of the HaloCam$_\mathrm{RAW}$ images, the measured signals range between 1000 and 3400 DN, where the mean deviation from a perfectly linear sensor amounts to 0.15%, 0.27%, 0.24%, and 0.04% for the R, G1, G2, and B-channel, respectively. For signals close to saturation (4095 DN) the sensor becomes strongly non-linear. Thus, signals $S_0 > 3400$ DN have to be excluded

from the analysis.

### 4.4 Spectral response

The spectral response of HaloCam$_\mathrm{RAW}$ was characterized in a similar way as described in Gege et al. (2009) and Baumgartner et al. (2012) using a collimated beam of the monochromator (Oriel MS257) at CHB. The monochromator has an absolute uncertainty of $\pm 0.1$ nm for $\lambda \leq 1000$ nm and $\pm 0.2$ nm for $\lambda > 1000$ nm with a spectral bandwidth of $0.65$ nm and $1.3$ nm,

respectively (Baumgartner, 2019). To keep the duration of the calibration procedure short, only a small region of $8 \times 8$ pixels (per channel) on the camera sensor was illuminated by the monochromator via a parabolic mirror. Measurements were performed over a wavelength range of $350$ nm to $900$ nm with steps of $5$ nm together with the window of the camera casing shown in Fig. 1. Figure 8 displays the result of the spectral calibration for the red, blue and the two green channels. To obtain the spectral sensitivity curves, the raw images were averaged over the illuminated pixel region and over a set of 10 images per wavelength.

Subsequently, the dark signal was subtracted and the spectral response for each channel was normalized to 1.

### 4.5 Absolute radiometric response

To obtain an estimate for the absolute radiometric response of HaloCam$_\mathrm{RAW}$ the images recorded on 22 September 2015 were cross-calibrated against simultaneous specMACS measurements. For five different specMACS scans the HaloCam$_\mathrm{RAW}$ images recorded within the time of the specMACS scan were selected and averaged. The absolute radiometric response of

HaloCam$_\mathrm{RAW}$ can be determined by dividing the normalized and flat-field corrected signal $s_\mathrm{n}$ in [DN/ms] by the specMACS



**Table 2.** HaloCam$_{\mathrm{RAW}}$ absolute radiometric response within a $1\sigma$ confidence interval.

| Channel | Radiometric response [DN ms$^{-1}$/(mW m$^{-2}$ nm$^{-1}$ sr$^{-1}$)] |
|---------|-----------------------------------------------------------------------|
| R  | $6.80 \pm 0.12$ |
| G1 | $5.79 \pm 0.14$ |
| G2 | $5.77 \pm 0.14$ |
| B  | $5.24 \pm 0.29$ |

radiance $L$ in $[\mathrm{mW\,m^{-2}\,nm^{-1}\,sr^{-1}}]$

$$R = s_{\mathrm{n}}/L. \tag{13}$$

An example of one of the five specMACS scans is displayed in Fig. 9 showing the upper part of a 22° halo. The first panel

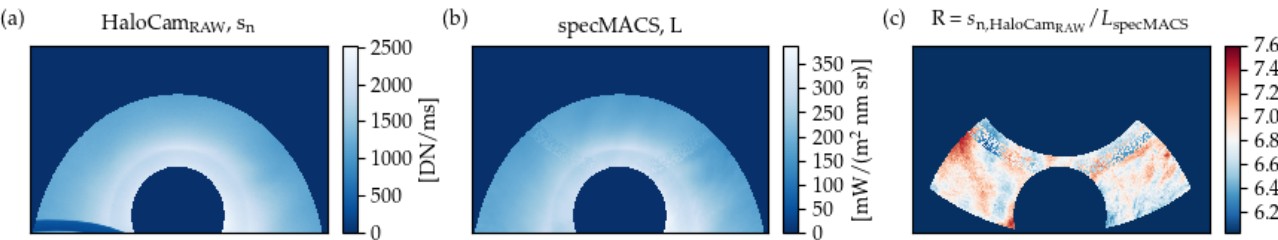

**Figure 9.** (a) HaloCam$_{\mathrm{RAW}}$ normalized and flat-field corrected signal $s_{\mathrm{n}}$ [DN/ms] for the R-channel (10:29:20 UTC) and (b) specMACS radiance $L$ weighted with the HaloCam$_{\mathrm{RAW}}$ spectral response of the R-channel showing the upper part of the 22° halo on 22 September 2015 (10:28:17 – 10:30:27 UTC). (c) The radiometric response was calculated by dividing $s_{\mathrm{n,HaloCam_{RAW}}}/L_{\mathrm{specMACS}}$, which were interpolated to the same angular grid. The image region was chosen to avoid the specMACS scan above the sun, which might be stray light contaminated, and the lower part of the HaloCam$_{\mathrm{RAW}}$ image, which happened to be obstructed by a cable in this instance.

(Fig. 9a) displays the normalized and flat-field corrected signal $s_{\mathrm{n}}$ of HaloCam$_{\mathrm{RAW}}$'s R-channel averaged within the time of

5  the specMACS scan from 10:28:17 to 10:30:27 UTC in Fig. 9b. The specMACS radiance was interpolated to the angular grid of HaloCam$_{\mathrm{RAW}}$ and weighted with the spectral response, here for the R-channel. The radiometric response for HaloCam$_{\mathrm{RAW}}$ is determined by dividing $s_{\mathrm{n}}$ from HaloCam$_{\mathrm{RAW}}$ by the specMACS radiance $L$, as depicted in Fig. 9c. Here, one radiometric response $R$ for all sensor pixels is determined by averaging over all pixels in Fig. 9c under the assumption that the photoresponse non-uniformity is already accounted for by the flat-field correction.

10  Table 2 provides the resulting radiometric response $R$ [DN ms$^{-1}$ mW$^{-1}$ m$^2$ nm sr] as defined in Ewald et al. (2016) averaged the five evaluated scenes. These values were derived using the specMACS scan to both sides of the sun as shown in Fig. 9c. The uncertainties are provided within a $1\sigma$ confidence interval and comprise specMACS's total radiometric uncertainty, which is computed for each pixel, and the standard deviation of the calibration factor calculated over all considered pixels.





## 4.6  Signal noise

The measurements of the LIS can also be used to estimate the noise $\mathcal{N}$ of the measured signal as described in Ewald et al. (2016). The noise consists of the dark noise $\mathcal{N}_{\mathrm{d}}$ and the photon shot noise $\mathcal{N}_{\mathrm{shot}}$. Thus, the standard deviation of the signal noise can be calculated by

$$\sigma_{\mathcal{N}} = \sqrt{\sigma_{\mathrm{shot}}^2 + \sigma_{\mathrm{d}}^2 + \sigma_{\mathrm{read}}^2}. \tag{14}$$

The number of photons $N$ detected over a time interval $t_{\mathrm{expos}}$ can be estimated by a Poisson distribution. A Poisson distribution with expectation value $N$ has a standard deviation of $\sigma_N \propto \sqrt{N}$. Thus, the variance of the photon shot noise $\sigma_{\mathrm{shot}}^2$ should scale linearly with the number of detected photoelectrons $N$ and the squared conversion gain $k^2$ [DN$^2$] and $\sigma_{\mathcal{N}}$ can be written as

$$\sigma_{\mathcal{N}} = \sqrt{k^2 N + \sigma_{\mathrm{d}}^2}. \tag{15}$$

Figure 10 shows a histogram of the variance $\sigma_{\mathcal{N}}^2$ (a) and the standard deviation $\sigma_{\mathcal{N}}$ (b) of the measured signal of the LIS.

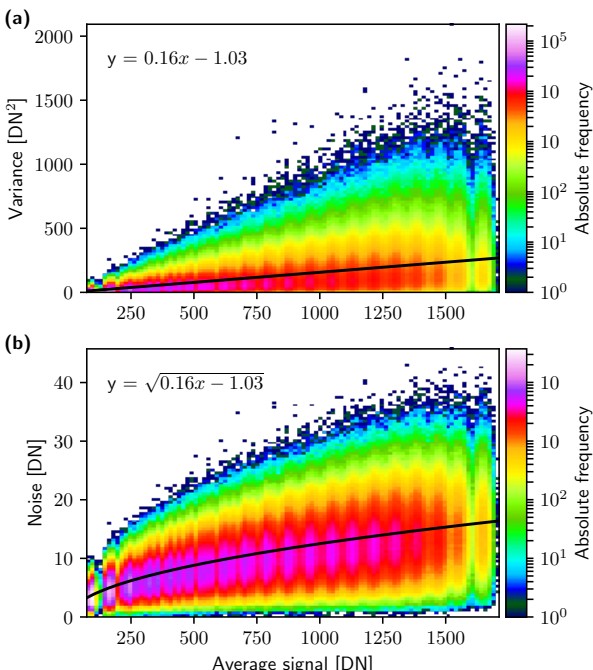

**Figure 10.** 2-dimensional histograms of the variance [DN$^2$] (a) and the noise [DN] (b) of the measured signal as a function of the averaged signal of the R-channel. The black lines are least-square fits, which are parameterized as indicated by the respective equation in the figure.

For this analysis each sensor pixel, evaluated over five images, was used for all exposure times (0.5 to $9.5\,\mathrm{ms}$). The results are shown for the R-channel here, but are very similar for the other three channels. According to Eq. (15), the variance of the signal measured by each pixel should scale linearly with the signal itself (Fig. 10a), whereas the standard deviation should





scale with the square root of the signal (Fig. 10b). In case the measurements deviate from the expected behavior, this would hint at possible non-linearities of the sensor. The signal noise ranges from about $10\,\mathrm{DN}$ for signals of about $100\,\mathrm{DN}$ to about $40\,\mathrm{DN}$ for signals of about $1500\,\mathrm{DN}$, typical for operational measurements.

### 4.7 HaloCam$_{\mathrm{RAW}}$ total radiometric uncertainty

**Table 3.** HaloCam$_{\mathrm{RAW}}$ total radiometric uncertainty with a $2\sigma$ confidence interval.

|  |  | Signal $S$ | R-channel | G1-channel | G2-channel | B-channel |
|---|---|---|---|---|---|---|
| **Relative radiometric uncertainty** | $\frac{2\sigma_{\mathrm{s_n}}}{s_{\mathrm{n}}}100\%$ | 1000 DN | 2.8 % | 2.4 % | 2.4 % | 3.3 % |
|  |  | 3000 DN | 1.8 % | 1.7 % | 1.7 % | 2.1 % |
| **Absolute radiometric uncertainty** | $\frac{2\sigma_{\mathrm{L}}}{L}100\%$ | 1000 DN | 5.0 % | 5.8 % | 5.8 % | 11.8 % |
|  |  | 3000 DN | 4.5 % | 5.5 % | 5.5 % | 11.5 % |

The total radiometric uncertainty of HaloCam$_{\mathrm{RAW}}$ was estimated by applying Gaussian error propagation to the equations describing the measured signal with the respective errors. Similar to the description in Ewald et al. (2016), the calculation of the total radiometric uncertainty will be outlined in the following. According to Eq. (1) the uncertainty of the radiometric signal $S_0$ is computed by combining the absolute uncertainties of the dark signal $\sigma_{\mathrm{d}}(t_{\mathrm{expos}}, T)$ and the instantaneous noise $\sigma_{\mathcal{N}}(S_0)$

$$\sigma_{\mathrm{S_0}} = \sqrt{\sigma_{\mathrm{d}}(t_{\mathrm{expos}}, T)^2 + \sigma_{\mathcal{N}}(S_0)^2}. \tag{16}$$

As defined by Eq. (12) the uncertainty of the normalized signal $s_{\mathrm{n}}$ consists of the relative uncertainty of the photoelectric signal $\sigma_{\mathrm{S_0}}$, the relative uncertainty of the flat-field calibration $\sigma_{\mathrm{F}}$, and the uncertainty due to the non-linearity of the sensor $\sigma_{\mathrm{nonlin}}$ according to

$$\frac{\sigma_{\mathrm{s_n}}}{s_{\mathrm{n}}} = \sqrt{\left(\frac{\sigma_{\mathrm{S_0}}}{S_0}\right)^2 + \left(\frac{\sigma_{\mathrm{F}}}{F}\right)^2 + \left(\frac{\sigma_{\mathrm{nonlin}}}{s_{\mathrm{n}}}\right)^2}. \tag{17}$$

Uncertainties due to polarization of light by components of the camera or the casing were not determined for HaloCam$_{\mathrm{RAW}}$.
However, according to Ewald et al. (2016), the largest part of the polarization sensitivity of specMACS is introduced by the transmission grating which adds the spectral dimension to the measurements. Since HaloCam$_{\mathrm{RAW}}$ is not equipped with a grating, it is assumed that its polarization sensitivity is significantly lower than for specMACS. Direct solar radiation is unpolarized and thus the degree of polarization for scattering angles in the region of the $22°$ halo is expected to be less than about 5% (Hansen and Travis, 1974; Emde et al., 2010). The degree of polarization for transmitted light in the region of the
$22°$ halo is lower than for observations of reflected light from cloud sides, especially in the rainbow scattering region, which is





the focus of Ewald et al. (2016). Thus, we can conclude that even if the polarization sensitivity of the camera was significant, the error in the measured signal would be very small due to the low degree of polarization of the incoming radiation. Therefore, the contribution of the polarization sensitivity to the total measurement uncertainty is considered negligible for $\text{HaloCam}_{\text{RAW}}$. Finally, the radiometric calibration accounts for the error of the sensor response $\sigma_{\text{R}}$, which was estimated from cross-calibration

between $\text{HaloCam}_{\text{RAW}}$ and specMACS

$$\frac{\sigma_{\text{L}}}{L} = \sqrt{\left(\frac{\sigma_{s_{\text{n}}}}{s_{\text{n}}}\right)^2 + \left(\frac{\sigma_{\text{R}}}{R}\right)^2}. \tag{18}$$

Table 3 provides the total relative and absolute radiometric uncertainties for the four channels of $\text{HaloCam}_{\text{RAW}}$ for two typical signals of $1000\,\text{DN}$ and $3000\,\text{DN}$. The relative radiometric uncertainty is an estimate of the error of the normalized signal $s_{\text{n}}$ (Eq. (17)) which is smaller than 4% for all four channels. For larger signals the relative $2\sigma$ uncertainty is smaller since

the absolute uncertainty is divided by a larger value (cf. Eq. (18)). This uncertainty is valid for signal ratios since they are independent of the sensor response $R$. For spectral radiance measurements, however, the uncertainty increases significantly due to the contribution of the uncertainty of the estimated sensor response $\sigma_{\text{R}}$. For the R-channel the total absolute uncertainty amounts to about 4.5% and about 5.5% for the two green channels. The uncertainty is largest for the B-channel with about 11.5%.

Since the radiometric response of $\text{HaloCam}_{\text{RAW}}$ was cross-calibrated against specMACS, its estimated uncertainty comprises both the relative radiometric uncertainty as well as specMACS's absolute radiometric uncertainty. Additional minor sources of uncertainty are: First, the $\text{HaloCam}_{\text{RAW}}$ images are recorded every $10\,\text{s}$, so the average temporal offset between the specMACS and $\text{HaloCam}_{\text{RAW}}$ measurements amounts to $5\,\text{s}$ which leads to small deviations due to cloud motion and inhomogeneities of the scene. Second, due to the temporal offset between the measurements the specMACS and $\text{HaloCam}_{\text{RAW}}$

scenes cannot be perfectly matched and a slight misalignment remains. Third, to compare the measurements, the specMACS observations have to be convolved with the spectral response of the four channels of $\text{HaloCam}_{\text{RAW}}$. For wavelengths at the edge of the spectral sensitivity of the specMACS sensor, the measurement uncertainty increases strongly introducing additional uncertainty in the estimated radiometric response for the $\text{HaloCam}_{\text{RAW}}$ measurements. This effect is responsible for the larger uncertainty of the blue channel, which has a spectral response centered at much shorter wavelengths. In this spectral region

specMACS has a larger measurement uncertainty compared to the red and green channels.

## 5 Application

With a radiometrically characterized camera it is possible to quantitatively analyze the measured radiance distribution. Figure 11a shows the R-channel of a $\text{HaloCam}_{\text{RAW}}$ image, averaged over 2 hours on 21 April 2016. The observations show a 22° halo with the direct sunlight blocked by the circular sun shade (cf. Section 2). The radiance at the 22° halo peak amounts

to about $220\,\text{mW}\,\text{m}^{-2}\,\text{nm}^{-1}\,\text{sr}^{-1}$ with a $2\sigma$ measurement uncertainty of about $10\,\text{mW}\,\text{m}^{-2}\,\text{nm}^{-1}\,\text{sr}^{-1}$ derived from the absolute radiometric characterization (cf. Fig. 11b). The 22° halo ratio (HR) serves as a measure of the brightness contrast of the halo display (e.g. Forster et al. (2017)) and amounts to about 1.03 in the azimuth segment indicated in yellow in Fig. 11a.


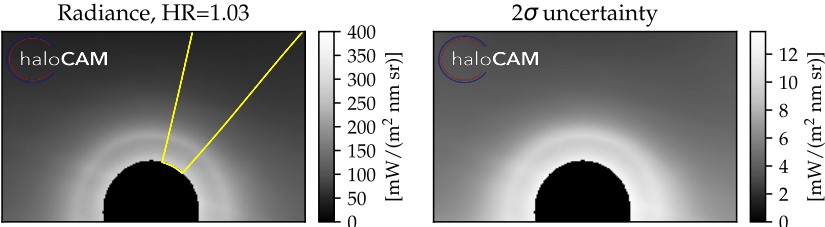

**Figure 11.** HaloCam$_{RAW}$ R channel (a) radiance and (b) $2\sigma$ standard deviation from 21 April 2016, averaged over the period between 12 and 14 UTC. The 22° halo ratio amounts to 1.03 and was calculated in the averaged azimuth segment, indicated here in yellow.

The measurement uncertainty of the 22° HR amounts to about 3% and is calculated by applying the relative radiometric characterization since a signal ratio is considered. Figure 12 displays radiative transfer simulations with libRadtran (Mayer and Kylling, 2005; Emde et al., 2016) using the DISORT solver (Stamnes et al., 1988) and ice crystal optical properties based on the parameterization of Yang et al. (2013). Three exemplary ice crystal habits were selected: plates (first row), hollow columns (second row), and solid columns (third row). The ice crystal optical properties of Yang et al. (2013) provide three degrees of surface roughness: smooth, moderately roughened, severely roughened. To allow for a more flexible parameterization of ice crystal surface roughness, simulations were performed for mixtures of severely roughened and smooth ice crystals as explained in Forster et al. (2017). This parameterization assumes that the cirrus cloud consists of two ice crystal populations: one population of smooth hexagonal crystals capable of forming a 22° and/or 46° halo and a second population consisting of severely roughened ice crystals which do not produce a halo display and serve as a "background" of scattering ice particles. These two populations are mixed by scaling their optical thickness according to their particle fraction in the cirrus cloud. Since the 22° halo is produced by smooth hexagonal ice crystals, the fraction of smooth ice crystals in this parameterization determines the HR, i.e. the brightness contrast of the 22° halo. The 22° HR in Fig. 12 increases from left to right with a growing fraction of smooth ice crystals. All remaining simulation parameters are kept constant (see Fig. 12 caption). To represent the HaloCam$_{RAW}$'s R-channel, libRadtran simulations were performed between 350 nm and 900 nm and averaged with the sensor's spectral response as shown in Fig. 8.

The simulated radiance values of the whole scene are comparable to the observations but the 22° and 46° HR varies significantly among the different ice crystal shapes and smooth crystal fractions (SCF). The radiative transfer simulations using solid columns produce the best matching radiance distribution compared to the HaloCam$_{RAW}$ observations. For a SCF between 20% and 30% they produce a 22° HR between 1.01 and 1.06. The simulations using ice crystal plates (Fig. 12 top) do not represent the observations well since they produce an additional 46° halo. Also hollow columns do not match the observations since the scattering angle region within the 22° halo close to the sun is much brighter compared to the observations. These findings are supported by Fig. 13: the simulation using ice crystal plates (blue) shows a small peak at the position of the 46° halo, which is not present in the observed radiance distribution (black). The simulation using hollow columns exhibits another peak at scattering angles of about 18° (orange line in Fig. 13). Qualitatively, the simulation using solid columns best represents the

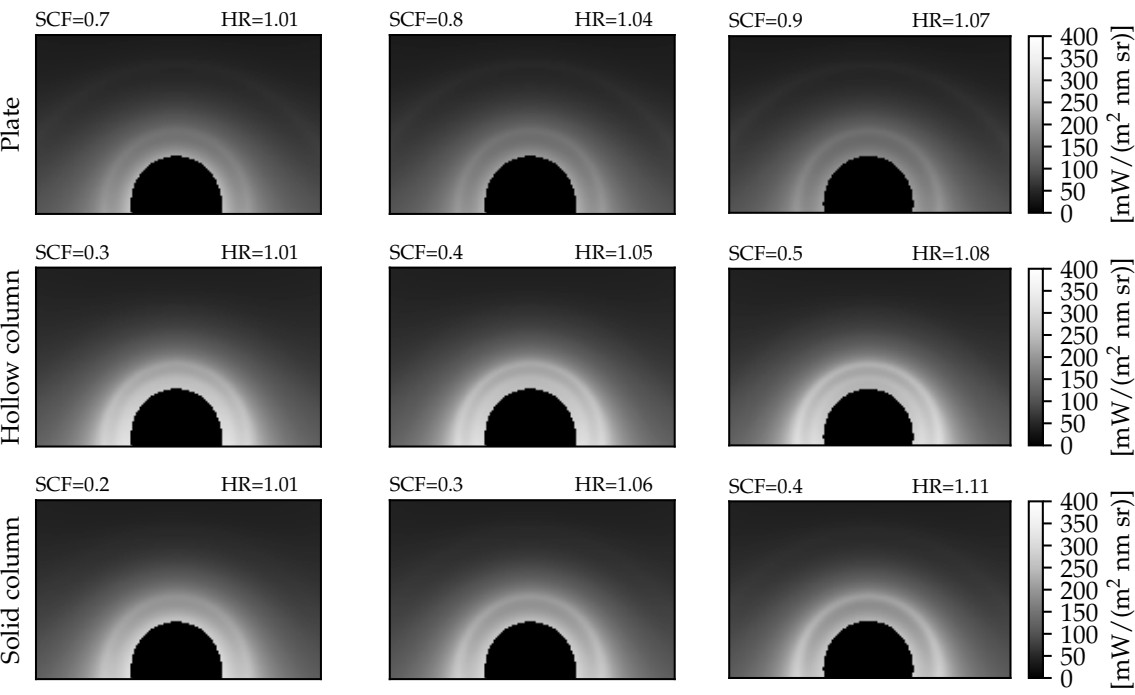

**Figure 12.** LibRadtran simulations using ice crystal plates (top), hollow columns (center), and solid columns (bottom) for a mixture of smooth and roughened crystals, with increasing smooth crystal fraction (SCF) from left to right. The SCF of the respective columns was chosen to achieve a similar 22° halo ratio (HR) for the three different habits. The observed HR lies between the simulated values of the left and center column. The remaining simulation parameters were kept constant: a cirrus optical thickness of 0.4 was chosen to roughly match the observed radiance values and an effective crystal radius $r_{\text{eff}} = 20\,\mu\text{m}$ as a typical value for cirrus clouds was assumed. The simulations were performed for an aerosol optical thickness 0.15 with the "continental average" optical properties mixture from OPAC (Hess et al., 1998), for an average solar zenith angle of SZA $= 42.8°$, assuming an absorbing surface, and using the spectral response of the red channel (cf. Fig. 8).





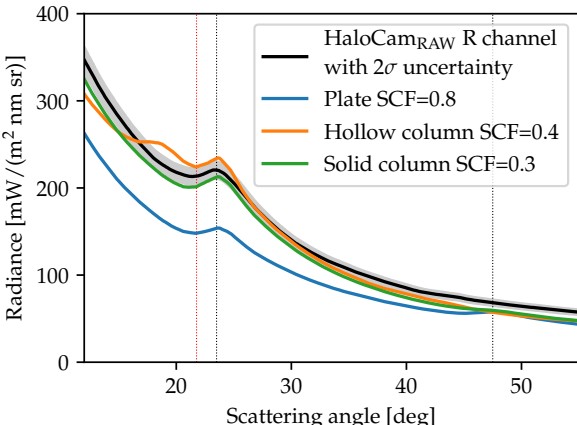

**Figure 13.** The black solid line displays the HaloCam$_{\mathrm{RAW}}$ radiance in the principal plane above the sun from the scene shown in Fig. 12a. The $2\sigma$ uncertainty of the HaloCam$_{\mathrm{RAW}}$ radiance is represented by the gray shading around the black line. The scattering angle of the 22° and 46° halo peaks are indicated by the vertical black dashed lines. The red dashed line indicates the minimum of the 22° halo. The 22° halo ratio is computed by the ratio between the radiance values of the maximum and the minimum. The colored lines represent the radiance in the principal plane from the libRadtran simulations shown in Fig. 12b (central column) for ice crystal plates (blue), hollow columns (orange), and solid columns (green).

observations. Since the cirrus and aerosol optical thickness as well as the ice crystal effective radius are only a rough estimate, a slight offset between observations and simulations remains. On this basis a method can be developed to retrieve ice crystal properties which best match the observations with help of radiative transfer simulations.

# 6 Conclusions

We present a procedure for both geometric and absolute radiometric characterization of HaloCam$_{\mathrm{RAW}}$. HaloCam$_{\mathrm{RAW}}$ is a camera designed for the quantitative analysis of halo displays and is part of the automated halo observation system HaloCam described in Forster et al. (2017). The geometric calibration was performed using a chessboard pattern with known dimensions to determine the camera matrix as well as the distortion coefficients of the RAW image with Bayer pattern. The sensor's dark signal was determined using a climate chamber in a dark room and with the camera lens covered. The photoresponse non-

uniformity (i.e. the vignetting effect), the spectral response of the RGB sensor, linearity of the sensor's radiometric response as well as signal noise were characterized at the Calibration Home Base (CHB) of the Remote Sensing Technology Institute at the German Aerospace Center in Oberpfaffenhofen. While the spectral response was characterized using a monochromator, the remaining effects were determined using the large integrating sphere (LIS) of the facility. The absolute radiometric response was estimated by cross-calibrating the HaloCam$_{\mathrm{RAW}}$ observations against the completely characterized specMACS imager

for simultaneously measured scenes of a 22° halo. Finally, the total radiometric uncertainty was determined by taking into



account the aforementioned error sources as well as the radiometric uncertainty of specMACS for the cross-calibration. The polarization sensitivity of HaloCam$_\text{RAW}$ was considered negligible.

For a typical measurement signal of 1000 DN the relative radiometric uncertainty amounts to 2.4% for both green channels, 2.8% for the red channel and 3.3% for the blue channel. For a larger signal of 3000 DN the relative radiometric uncertainty

ranges between 1.7% for the green channels and 2.1% blue channel. The absolute radiometric uncertainty is larger due to the additional uncertainty of specMACS and amounts to 5.0% for the red channel, 5.8% for the green channels and 11.8% for the blue channel for a sensor signal of 1000 DN and is similar to a signal of 3000 DN since a large part of the additional uncertainty arises from the inhomogeneity of the scene used for cross-calibration and the absolute radiometric uncertainty of specMACS.

The geometric and radiometric characterization of HaloCam$_\text{RAW}$ were applied to a scene observed on 21 April 2016 when a

22° halo was present for about 2 hours. The observed radiance distribution was compared to radiative transfer simulations using solid columns, hollow columns and plates as well as three different mixtures of severely roughened and smooth ice crystals which produced a similar 22° halo ratio as the measurements. The remaining parameters of the cirrus were kept constant: an optical thickness of 0.4 was chosen to roughly match the absolute values of the HaloCam$_\text{RAW}$ radiances and an effective crystal radius of 20 μm was assumed, which is a typical value for cirrus clouds. Although this parameter choice produces a comparable

brightness contrast for the 22° halo, some ice crystal habits produce additional features in their radiance distribution, which are not visible in the HaloCam$_\text{RAW}$ observations and can be excluded in this case. For example, plates produce an additional 46° halo and hollow columns feature another the radiance peak at a scattering angle of about 18°. This comparison demonstrates the potential for developing a retrieval method for ice crystal properties including shape and roughness.

The absolute radiometric characterization allows to compare HaloCam$_\text{RAW}$ observations with radiative transfer simulations.

If the HaloCam$_\text{RAW}$ observations are analyzed using ratios of radiance values, the relative radiometric uncertainty applies. This characterization is an important pre-requisite to develop a quantitative retrieval of ice crystal properties, like crystal shape, size, and roughness, from observations of halo displays using HaloCam$_\text{RAW}$. Using a long-term database of calibrated HaloCam$_\text{RAW}$ observations together with radiative transfer simulations typical ice crystal properties of halo-producing cirrus clouds can be retrieved which will be the focus of future publications. The retrieved ice crystal properties have the potential to

complement space-borne retrievals by adding information about the forward scattering part of the phase function.

*Data availability.* The HaloCam$_\text{RAW}$ data used for the camera characterization and the simulation results will be provided upon request.

*Author contributions.* LF and AB performed the measurements for the characterization of HaloCam$_\text{RAW}$ at CHB, DLR. LF evaluated the measurements and performed the radiometric and geometric characterization of the camera. LF also performed the radiative transfer simulations and evaluation of the case study for demonstrating the application of the camera characterization. MS designed and manufactured the

weather-proof housing of the camera. TK assisted with the specMACS measurements and data processing for the absolute calibration. BM provided valuable input during the development of the methods and for the final version of the manuscript.





*Competing interests.* The authors declare that they have no conflict of interest.

*Acknowledgements.* We would like to thank Hans Grob for providing the software for controlling the Weiss climate chamber.



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
