# Peer review of "Ice Crystal Characterization in Cirrus Clouds II: Radiometric Characterization of HaloCam for the Quantitative Analysis of Halo Displays"

_Atmospheric Measurement Techniques, 2019_

## Referee Comment (RC1) · Anonymous Referee #1 · 22 Mar 2020

This paper describes the radiometric calibration of a surface-based camera designed to detect and characterize halos near the solar disk caused by ice crystals in clouds. It accompanies the early paper by Forster et al. (2017) that introduced a version of the camera. It is a highly technical paper, but also includes a short section on the application of the calibrated halo camera. The paper is very well written. The figures are very clear. I have only two minor suggestions for improvement, as described below. The paper can be accepted after addressing these comments.

1) The introduction is a bit short and lacks a good motivation for such a camera. Please elaborate more on why a calibrated halo camera is useful for cloud, atmospheric and

climate physics.

2) The application described in section 5 assumes randomly oriented crystals only. Please state this in the text. Please add some discussion on how this type of analysis would possibly be affected by the presence of oriented crystals. In particular, parts of the supralateral arc by oriented columns are very close to the 46-degree halo. Thus, the analysis might be biased if a supralateral arc is present instead of the more rare 46-degree halo. The distinction between the 46-dgeree halo and supralateral arc is discussed on this website: https://www.atoptics.co.uk/halo/46orsup.htm Moreover, I am wondering whether the camera has the resolution and accuracy to, in principle, detect and distinguish other arcs caused by oriented crystals (many are described on the atoptics website).
* * *

---

## Referee Comment (RC2) · Anonymous Referee #2 · 22 Apr 2020

General Comments:

The submitted manuscript deals with the geometric and radiometric calibration of a camera system HaloCamRAW designed for halo tracking. The long-term objective is to perform a quantitative analysis of ice crystal properties in cirrus clouds from the halo database provided by this camera combined to a radiative transfer model. The article explores prerequisite steps to achieve this goal through one halo case.

The article is well written and easy to follow through the different sections and figures. It is mainly dedicated to the radiometric calibration procedure. Indeed, the methodology for geometric calibration with chessboard checkers (Zhang 2000) is widely used and

referenced. The radiometric calibration consists in 2 steps: a signal correction following the same steps as used in (Ewald et al. 2016) for the specMACS imager, and a cross calibration step using specMACS imager to estimate the radiance measurement from the camera signal. The paper includes a theoretical uncertainty analysis and gives in details the assumptions that are made in the calibration process. In the last part, one application is performed on a 22° halo case. The authors perform several simulations of LibRadtran model with various ice crystals/cirrus cloud parameters inputs. The results show that one combination of parameters gives a good agreement with the HaloCamRAW estimated radiance. Under certain assumptions this methodology allows to characterize some parameters of ice crystals in the cirrus cloud.

Specific Comments:

1) More context would help the reader to appreciate the value of this work. Indeed, the global scope of this work should be given in introduction. Advantages (and eventually drawbacks) of ice crystal characterization in cirrus clouds by the HaloCam system compared to others instruments could also be given.

2) Line 2 p.19, the authors write "a method could be developped to retrieve ice crystal properties". Is that method supposed to be automated to make a "long-term database" (line 22 p.20) ? In that case, what are the difficulties to overcome for this purpose ? It would be interesting to discuss about that point. For instance, halo cases with quite different sets of parameters might lead to the same radiance response using the libRadtran model, hence the inversion of the operator will not be possible if no additional information or constraint is added.

---

## Author Comment (AC1) · 20 May 2020

**Reply to comments by referee #1**

We thank the referee for carefully reviewing the manuscript and for the valuable suggestions and comments, which are addressed below. The referee's comments are highlighted in blue.

**1) The introduction is a bit short and lacks a good motivation for such a camera. Please elaborate more on why a calibrated halo camera is useful for cloud, atmospheric and climate physics.**

We appreciate your suggestion and expanded further on the motivation for a calibrated camera for halo observations. Referee #2 raised a similar concern. We extended/re-wrote the introduction as follows:

[revised manuscript text omitted]

Please add some discussion on how this type of analysis would possibly be affected by the presence of oriented crystals. In particular, parts of the supralateral arc by oriented columns are very close to the 46-degree halo. Thus, the analysis might be biased if a supralateral arc is present instead of the more rare 46-degree halo. The distinction between the 46-dgeree halo and supralateral arc is discussed on this website: https://www.atoptics.co.uk/halo/46orsup.htm.

The referee is raising a very important point. We added more discussion about the possibility to retrieve information about oriented ice crystals and the possible overlap of halo displays formed by oriented and randomly oriented crystals. The main focus of the present study is on the characterization of  $HaloCam_{RAW}$  including a simple demonstration assuming randomly oriented crystals. The actual development of a quantitative retrieval method will be the focus of a future study, where it will also be necessary to discuss the separate treatment of halo displays formed by randomly oriented and oriented crystals. We added the following paragraph at the end of Section 5:

"While this study focuses solely on randomly oriented crystals, halo displays caused by oriented crystals, such as sundogs, upper tangent arcs as well as the rare Parry and supralateral arc (e.g. Greenler, 1980; Tape, 1994) can also be observed within HaloCamRAW's FOV and contain important information about the fraction of oriented plates and columns. Depending on the solar elevation, halo displays formed by oriented and randomly oriented crystals overlap in certain image regions, as for example upper tangent arc and  $22^{\circ}$  halo or supralateral arc and  $46^{\circ}$  halo (Cowley). For the future development of a quantitative retrieval method, care must be taken to treat those image regions appropriately."

Moreover, I am wondering whether the camera has the resolution and accuracy to, in principle, detect and distinguish other arcs caused by oriented crystals (many are described on the atoptics website).

In principle, HaloCam can distinguish between features with an inter-pixel resolution of  $0.1^{\circ}$  (for each color channel with 608x968 pixels; p. 6, l. 12) and a pointing accuracy of about  $0.5^{\circ}$  (p. 4, l. 25). This is a very interesting question

and motivates an in-depth discussion about the analysis of more infrequently observed halo displays caused by oriented crystals such as e.g. Parry arcs. Since the application in this study only focuses on randomly oriented crystals, we will leave this discussion for a future publication but added a short sentence on p. 20, 1. 14:

"Halo displays caused by oriented ice crystals, such as sundogs, upper tangent arcs as well as the rare Parry and supralateral arc (e.g. Greenler, 1980; Tape, 1994), could also be observed within  $HaloCam_{RAW}$ 's FOV and add important information about the fraction of oriented columns and plates."

**References**

[revised manuscript text omitted]

---

## Author Comment (AC2) · 20 May 2020

**Reply to comments by referee #2**

We thank the referee for carefully reviewing the manuscript and for the valuable suggestions and comments, which are addressed below. The referee's comments are highlighted in blue.

1) More context would help the reader to appreciate the value of this work. Indeed, the global scope of this work should be given in introduction. Advantages (and eventually drawbacks) of ice crystal characterization in cirrus clouds by the HaloCam system compared to others instruments could also be given.

We appreciate your suggestion and expanded further on the motivation for a calibrated camera for halo observations and comparison with other instruments. Referee #1 raised a similar concern. We extended/re-wrote the introduction as follows:

[revised manuscript text omitted]

2) Line 2 p.19, the authors write "a method could be developped to retrieve ice crystal properties". Is that method supposed to be automated to make a "long-term database" (line 22 p.20) ? In that case, what are the difficulties to overcome for this purpose? It would be interesting to discuss about that point. For instance, halo cases with quite different sets of parameters might lead to the same radiance response using the libRadtran model, hence the inversion of the operator will not be possible if no additional information or constraint is added.

Thank you for addressing this important point. We added some more discussion at the end of Section 5:

[revised manuscript text omitted]